# A Primer for Design and Systems Thinkers: A First-Year Engineering Course for Mindset Development

Jac Ka Lok Leung [1,*] 🔘, Davy Tsz Kit Ng [2] 🔘 and Chi-Ying Tsui [1]

1   Division of Integrative Systems and Design, The Hong Kong University of Science and Technology, Clear Water Bay, Hong Kong; eetsui@ust.hk
2   Faculty of Education, The University of Hong Kong, Pok Fu Lam, Hong Kong; davyngtk@connect.hku.hk
*   Correspondence: egjac@ust.hk

**Abstract:** Teaching students to think in complex systems and design is presumably intricate, creative, and nonlinear. However, due to the overwhelming number of standardized tools and frameworks, the process sometimes ends up being procedural and deductive. Conformity to rigid procedures loses the intention of creative problem-solving towards tackling wicked problems. This paper proposes a project-based approach to instill the mindsets for those who aspire to be design and systems thinkers through a first-year engineering course. Using the ADDIE model, the instructional design was implemented in three modules focusing on design, systems, and integration for real-world applications. The instructional design was evaluated via course feedback surveys and focus-group interviews. Students indicated positive impacts on creative mindsets, habits of systems thinkers, and interdisciplinary awareness. However, negative comments about the course arrangement such as heavy workload and disconnection between topics were identified. Suggestions from students, challenges faced by the instructors, and recommended practices are discussed. In times of increasing need to reform higher education due to digitization and artificial intelligence, this study provides a timely investigation of a new project-based and mindset-focused pedagogy in design and systems thinking education.

**Keywords:** creative mindset; design thinking; interdisciplinary studies; project-based learning; engineering education





## 1. Introduction

*"Today's problems come from yesterday's solutions."*—*Peter Senge.*

Designers and engineers have long played the role of using creative and technological methods to design and build for the betterment of humankind [1]. However, disruptive innovations are also increasing the complexity and the pace of today's world. The integration of expertise from various disciplines has become increasingly important in our interconnected society, and the ability to think integratively is a crucial skill for success in emerging careers [2,3]. Addressing problems that arise in the form of intricate and dynamic systems require more than mathematical deduction, i.e., to be undertaken via systemic, creative, and humanistic approaches.

To this end, design thinking and systems thinking have gained popularity among educators as approaches for teaching students to think outside of the box and tackle complex problems [4]. Despite the open-ended nature of these approaches, learning modules commonly found today, especially in crash courses, would focus on the standardized steps (e.g., design thinking process—empathy, define, ideate, prototype, test) and tools (e.g., empathy map, user journey map, etc.) rather than the values and mindsets of design and systems thinking. After these short modules, students may only recall the steps and tools and find difficulty in conceptualizing these "thinkings" and adapting to other contexts. Recently, critics have pointed out the pitfalls of this routine and "over-promising" process

and argued that "the shine of design thinking has been wearing off" [5]. Another criticism claimed that the process has often become overly simplistic or formulaic [6].

To address these concerns, this paper proposes a first-year projects course to instill creative mindsets at the beginning of students' design and systems thinking curriculum to help students unlearn some of the traditional perceptions of linear and siloed thinking, as well as to serve as a primer to prepare students for tackling open-ended scenarios as they progress through their studies. The course was newly introduced to complement the existing integrative systems and design curriculum.

Grounded in an experiential learning and constructivist approach, students developed their mindsets through three phases focusing on different areas—design, systems, and integration for real-world applications. The instructional design was evaluated via course feedback surveys and focus-group interviews. The evaluation focused on assessing students' mindsets' development. These mindsets include creative mindsets [7], habits of systems thinkers [8], and interdisciplinary awareness [9]. Explanations as to why these mindsets are important to design and systems thinkers are described in the next section.

Overall, the course was positively received. High engagement in class activities was observed. Students were motivated in all three projects and perceived to have improved their creative thinking, systems thinking, and other transferable skills. Students expressed feeling a lack of connection between projects. Some students were concerned about the assignment workload. As part of a design-based research roadmap, more work is needed to elicit a rigorous method to measure changes in students' mindsets.

The significance of this study is threefold. Firstly, few studies of design thinking pedagogies center on nurturing students' mindsets for integrative systems and design. Secondly, this study explores an alternative approach to designing learning activities for first-year engineering students in a design thinking course. Thirdly, this study provides empirical information on students' development in terms of their values and mindsets towards design and systems thinking.

## 2. Literature Review

### 2.1. Design Thinking Approaches and Human-Centered Designs

When problems arise in the form of complex and dynamic systems, traditional approaches to thinking linearly about problem-solutions or using singular cause-and-effect logic are no longer applicable to addressing the real needs of people. Take the case of the global pandemic COVID-19 as an example. Although the fundamental root cause is the virus, the problems that arise are more far-reaching than the virus itself; solutions span across technological, social, economic, and political domains and would not be effective without careful and holistic considerations of their multitudinous impacts. Any innovation in one system may trigger a string of backfires in the others. Products, services, and systems nowadays should be designed via systemic, creative, and humanistic approaches.

To this end, new education programs are advocating the notion of design thinking approaches and human-centered designs. Design thinking is taught as a problem-solving methodology that emphasizes empathy, experimentation, and collaboration [4]. The most credited organization to first conceptualize design-thinking was IDEO, founded by Stanford University professor David Kelley and then popularized by the Hasso Plattner Institute of Design (or d.school) at Stanford University (Stanford d.school: https://dschool.stanford.edu/ (accessed on 22 November 2023)). The d.school has developed a curriculum around design-thinking concepts that are widely used in universities, corporations, and organizations worldwide [10]. To date, design thinking has gained a significant foothold in academia, and its adoption continues to grow as more universities recognize its value in preparing students for the complex challenges of the 21st century. Besides IDEO, a few examples of adopting the design-thinking educational programs are as follows: in the US, the innovation of products and services, i.e., MIT's approach to design thinking at MIT Sloan (Innovation of Products and Services: MIT's Approach to Design Thinking: https://admissions.emeritus.org/programs/mit_sloan_executive_education/

design-thinking (accessed on 22 November 2023)); in Denmark, the design and innovation program at the Technical University of Denmark (Design and Innovation Program at DTU: https://designthinking.dtu.dk/english/ (accessed on 22 November 2023)); in Japan, the Master's program offered by Keio University's Graduate School of Media Design (Design Thinking and Innovation program at Keio: https://www.kmd.keio.ac.jp/academics (accessed on 22 November 2023)); and in Hong Kong, the division of integrative systems and design at the Hong Kong University of Science and Technology (Integrative Systems and Design at HKUST: https://isd.hkust.edu.hk/about-isd#what-is-isd (accessed on 22 November 2023)).

These programs all place an emphasis on human-centered design and the application of design- or systems-thinking processes for systematic innovations. This approach usually includes several steps to unravel the user's real needs and design solutions around them. It offers a great way to promote collaboration and creation in a wide range of contexts, and the notion of hands-on prototyping also helps students to construct physical products or services that redefine systems to enhance user desirability.

This approach gained popularity as it appears to guarantee results from students, offering some creative ways of addressing a problem. However, critics have challenged this claim as "over-promising" [5]. Ackermann's article posits that "... its short-term focus on novel and naive ideas has resulted in unrealistic and ungrounded recommendations". Another criticism is that the process would often be overly simplistic or formulaic and may not fully engage the creative and critical thinking skills of designers and innovators [11].

Indeed, when students are presented with many steps and tools, they would feel natural to follow these steps as if they are checking a series of boxes as part of their assessment. In particular, for engineering students, the majority of their training is to apply equations and generate well-defined models as solutions [12]. Introducing the concepts of open-mindedness and interconnectedness with no direct or right-or-wrong answers to wicked problems may sometimes be daunting [13]. As such, teaching students to think in systems and design should not only focus on the steps and tools but also, more importantly, on nurturing their mindsets to embrace non-linearity and divergent thinking.

*2.2. Foundation for Instructional Design*

In search of an epistemological foundation for instructional design, experiential learning theory (ELT) by David Kolb [14] and constructivism by Jean Piaget [15] are commonly cited references for project-based environments.

ELT emphasizes the importance of learning through direct experience and reflection. It consists of four stages: concrete experience, reflective observation, abstract conceptualization, and active experimentation [14]. ELT suggests that learners learn by actively engaging in real-world experiences and reflecting on their observations. Research findings indicate that incorporating ELT in the instructional design of project-based engineering courses leads to several advantages [16,17]. Firstly, it promotes active and hands-on learning, enabling students to apply concepts to practical situations. Second, ELT emphasizes problem-solving skills development by providing opportunities for students to reflect on their experiences and identify patterns, systems and connections. Lastly, ELT fosters a deeper understanding of the subject matter and enhances knowledge retention through active experimentation and application.

Constructivism posits that learners construct knowledge through their experiences and interactions with the environment. Learners assimilate new information by incorporating it into their existing mental structures and accommodate these structures when faced with new experiences that challenge their understanding [15]. Research suggests that incorporating constructivist principles into instructional design enhances students' motivation, engagement, and knowledge acquisition [18,19].

Both ELT and constructivism advocate learner-centered instructional design. They encourage instructors to create a collaborative and interactive learning environment where students can construct their own understanding from their learning experiences. By

actively participating in project-based activities, students develop a deeper understanding of engineering principles and their applications.

### 2.3. Mindsets for Integrative Systems and Design

Conformity to rigid procedures loses the intention of creative problem-solving for tackling wicked problems. Merely recalling standardized steps and tools would be unlikely to lead to success in coming up with out-of-the-box solutions to a problem. Instead, nurturing creative mindsets can drive one's creative behavior, and these behaviors will steer us toward creative achievements and innovations [7]. "A creative mindset can be a powerful force for looking beyond the status quo" (p. 24) [20]. As Karwowski and colleagues explained, creative mindsets are the continuum of a fixed-to-growth mindset and stability-to-changeability [9]. To nurture creative mindsets, Tom and David Kelly at IDEO suggested that it is important to inspire curiosity and exploration, allow for open-ended play and experimentation, and provide opportunities to try new things and make mistakes without judgment or criticism [20].

Breaking down complex problems into manageable pieces requires systems thinking. Fourteen "habits of a systems thinker" were proposed by Waters Center for Systems Thinking [21]. Examples of these habits include seeing the big picture, changing perspectives, surfacing, testing assumptions, etc. Developing the habits of a systems thinker would help students to understand how systems work, how subsystems are connected, and how actions taken can induce changes seen over time [8].

Interdisciplinary awareness, also referred to as interdisciplinary attitudes, is crucial for learning integrative systems and design, as it allows for a more comprehensive understanding of real-world problems and the development of innovative solutions. According to Klein [22], interdisciplinary studies bring together diverse perspectives, methods, and knowledge from various disciplines to address interconnected issues that cannot be effectively tackled through a single disciplinary approach. The awareness of interdisciplinarity is being able to see the value when working in teams with students in other disciplines.

### 3. Methods

To facilitate a structured and holistic instructional design process, this study adopts the "ADDIE" instructional design model. ADDIE is the acronym used to describe the five phases of an instructional design process: analysis, design, development, implementation, and evaluation [23,24]. The analysis, design, and development processes are described in Section 4 (Course Design), whereas the implementation and evaluation are reported in Section 5 (Results).

### 3.1. Research Design

This study used an action research methodology. Action research, or practitioner-based research, is a systematic and reflective inquiry conducted by individuals or groups, with the aim of improving their own practices and generating practical knowledge [25]. According to Jean McNiff, action research emphasizes the importance of practitioners engaging in a cyclical process of planning, acting, observing, and reflecting in order to enact and evaluate changes in their practice. One of its key advantages is the active involvement of practitioners, such as instructors, in the research process. The role of an instructor–researcher has a unique advantage in understanding the intricacies of the educational environment and the challenges students face, which can inform the research process and the development of interventions [26].

To address potential biases in data collection and student assessments, several measures were implemented in this study. Firstly, data were collected from multiple sources, including quantitative course feedback and qualitative focus-group interviews. By triangulating data from diverse sources, the researchers can capture a holistic understanding of students' development. Secondly, pre-established questionnaire and interview protocols were used to maintain consistency and minimize bias. Thirdly, the results and findings were

verified and agreed by the authors and participants. For authors, two of them assumed the role of instructor–researcher and one was a researcher from another institution. For participants, the transcribed data were blinded and sent back to verify for any discrepancies.

### 3.2. Participants

The course named "Introduction to Integrative Systems and Design (course code: ISDN1001)" was offered for its first implementation. Thirty-two students enrolled. There were twenty-one male and eleven female students. All participants were first-year engineering students. The first and second authors were both instructors of this course. The course was newly introduced as a core course for students studying on the integrative systems and design (ISD) program. The ISD program was established in 2018 at the school of engineering to promotes experiential project-based learning. The development of course syllabus and activities were consulted by faculty members in ISD.

### 3.3. Data Collection

Since this course was newly introduced into the curriculum and the course objectives and content were notably different from those of other existing courses, this study adopted a "pre-experimental one-group post-test only design" (p. 323) [27]. It is a type of quasi-experiment in which the outcome of interest is measured after exposing a group of participants to the intervention. Due to its limitation for comparison, the findings will focus on the feasibility of the intervention and qualitative feedback and suggestions from the participants.

Students' learning experience was evaluated via course feedback surveys and semi-structured focus-group interviews. Two surveys were administered. One was the student feedback questionnaire, which was opened to students in the last two weeks of the semester. It consisted of eight 5-point Likert-scaled items and one open-ended question. Students gave ratings to the generic aspects of the course and the course instructors (see Appendix A). The other was conducted as a reflection exercise on the last day of class. Students used Mentimeter (Mentimeter: https://www.mentimeter.com/ (accessed on 22 November 2023)) to provide anonymous responses to four questions related to their mindset development (see Appendix A).

Subsequently, students were invited to focus-group interviews conducted by the first author. The interviews aimed to follow up on responses from the course feedback surveys to better understand each student's learning experiences with this new pedagogy. Eleven participants agreed to attend the focus-group interviews, and they were separated into three groups, each consisting of three to four participants. The interviews took place after students had received their course grades.

Prior to any distribution of surveys or invitation to focus-group interviews, students were informed about the purpose and procedure of the research. They were informed that the entire process is voluntary, which means they could withdraw from the surveys or interviews at any point without any consequences. In addition, students were ensured that all responses would be kept anonymous and all faces in photos would be censored for confidentiality.

## 4. Course Design

### 4.1. Analysis

The analysis stage includes identifying the instructional problem and defining the instructional gap [23,28]. Students taking this course are first-year engineering students. As most of the incoming high school students do not have any background in the design or knowledge of integrative systems, this course serves to let students understand the impact of integrative systems and the importance of design in creating these systems. As an introductory course, it also aims to introduce students to active learning elements, including hands-on, real-world application, project-based, and team-based learning experiences.

The intended learning outcomes include the following goals:

- Identify the impact of design and integrative systems in social, economic, and cultural contexts;
- Appreciate the values of integrative systems and design;
- Connect different disciplines in integrative systems and design;
- Recognize the value of and identify the interaction between different building blocks of systems;
- Develop the ability to work within constraints (e.g., technology boundary) and prioritize needs;
- Integrate knowledge from integrative systems.

### 4.2. Design and Development

The design stage includes identifying and creating overarching goals, assessment deliverables, instructional strategies, and media [23,28]. Corresponding to the learning theories described in the previous section, the content is roughly arranged into an introduction, followed by three modules: (1) design, (2) systems, and (3) integration for real-world applications (see Table 1). Each module involves a mini-project to exemplify their learning.

**Table 1.** Summary of the course design.

| Week | Module | Referenced Literature | Topics | Mini-Projects |
|------|--------|----------------------|--------|---------------|
| 1–3 | Introduction | [29,30] | Good and bad designs, creative thinking, divergent thinking | |
| 4–6 | Design | [20] | Social innovation, storytelling, design styles, interaction design | Student–staff observation trip |
| 7–9 | Systems | [8] | Systems design, systems and functions, the role of technology | Product disassembling and reassembling |
| 10–13 | Integration for real-world applications | [31] | | Smart city solutions |

The introduction helps to set the tone for the course. Instructors explain that the core objective of this course is to help to shift students' mindsets from close-ended to open-ended, from textbook to hands-on, from teacher-led to student-led, and from exam-based to project-based. References to create class activities were adapted from the Torrance Test of Creativity Thinking (TTCT) [29] and the role of latent cognitive processes in divergent thinking [30].

The "design" module aims to let students learn from shadowing more experienced teaching staff in a collaborative and open environment to develop their design and creative mindsets. The learning activities in this module follow the key principles proposed by Kelly and Kelly [20] that could promote the development of individuals' creative potential:

- Embrace empathy: understand others' needs and desires for innovative solutions;
- Embrace ambiguity: embrace uncertainty to explore new possibilities;
- Embrace failure: learn from failures and see them as valuable opportunities;
- Foster collaboration: leverage diverse perspectives for collective intelligence;
- Encourage divergent thinking: explore multiple perspectives for innovative ideas;
- Practice reframing: challenge assumptions and uncover new insights;
- Cultivate a bias towards action: prototype and iterate ideas for continuous improvement;
- Foster a growth mindset: believe in the ability to learn and develop new skills.

The "systems" module instills the concepts of physical and abstract systems and subsystems. This module involves basic ideas of a system and its classification, before engaging students with hands-on practices. This module utilizes a maker space as the learning environment to allow students to create meanings from artifacts. According to Waters Center for Systems Thinking (see Section 2.3), the learning activities in this module were designed to develop the "habits of a systems thinker" in the following ways [8]:

- Recognize interconnections: identify the dependencies and relationships between system elements;
- Think in systems: take a holistic approach, considering the entire system rather than isolated parts;
- Define system boundaries: determine the scope and context of the system under study;
- Foster continuous learning: embrace a mindset of ongoing improvement and adaptation.

The final phase serves two main purposes. Firstly, students are guided to reflect on and demonstrate what they have learned in previous weeks. Secondly, instructors would elucidate students' mindset development and help to transfer them to future contexts. A mini-project about "integration for real-world applications" is designed to raise their interdisciplinary awareness. In a team project format, students determine the project scope and distribute each member to an area for research. They share findings and collectively create a systemic design as their proposed solution. By incorporating insights from multiple fields, students can develop a more holistic understanding of the system they are studying and better identify the underlying causes and potential solutions to the challenges they face [31].

## 5. Results

### 5.1. Implementation

Classes were conducted in an interactive learning environment. Students sat in teams of four. Unlike typical group project courses, new teams were formed in each module. This decision was based on several considerations. Firstly, maintaining the diversity of student backgrounds in each team could foster creative mindsets with diverse perspectives. Secondly, the introductory nature of this course facilitated opportunities for students to meet and work with more students and faculty. Thirdly, every time a new team was introduced, the instructors would conduct a short icebreaker for team building and highlight the importance of learning to work in interdisciplinary teams.

The majority of the topics listed in Table 1 followed the same structure: introduction, reflective questions, and learning activities related to the module. Guest speakers from the faculty were invited to give short introductions to the topics. Then, reflective questions were asked to prompt students for anonymous responses through an online platform named Mentimeter. These questions were asked at the beginning of each class, so as to arouse students' interest in the topics, as well as to connect to the speakers' content. After that, the instructors guided students to make use of their newly acquired knowledge to work on tasks related to the modules.

The topics in the first three weeks included how designs are judged, human-centered design, dimensions of divergent thinking, and creative thinking. Class activities highlighted the practice of being sensitive to their surroundings and developing a habit of imagining "what-ifs". One of the out-of-class activities involved compiling a list of design observations. Students were asked to document what they feel are clever or designs with which they feel annoyed. Through this exercise, students understood some basic principles of good design while acknowledging that there is no absolute answer. In the first few weeks of in-class activities, students went through a series of divergent thinking and creativity exercises. Verbal and figural tests, such as "guessing consequence" and "picture completion", were adapted from TTCT [29]. Each exercise represented the training on different dimensions, such as flexibility, elaborateness, originality, and fluency.

Topics in the design module included social innovation, storytelling, design styles, and interaction design. This module revolved around a mini-project about a "student-staff observation trip" in the city. Students, in teams of four, discussed and planned a 2-h route in the city with designated teaching staff. During the trip, the staff shared their insights about interesting objects, interactions, systems, architecture, and even atmosphere with the students to exemplify the appreciation of design. Students took notes during the trip and used storytelling to share their observations with the class. The instructors reminded students that there is no definite answer to this exercise and they should try to be as creative as possible in the process. Students were encouraged to empathize with the characters in their story and try to convey those experiences to their audience. In teams, they presented highlights and insights from the trip. Also, they created a team journal that included a storyboard to highlight thought-provoking designs they encountered. The media of the presentation and team journals were flexible. They used narratives, animations, role-play, and storybooks.

The topics in the systems module included systems design, systems and functions, and the role of technology. The mini-project was about disassembling and reassembling a physical product. Each team received a recycled electronic device (e.g., a portable Bluetooth speaker, a dashboard camera, an electric heater, etc.). Before disassembling the device, students created a system breakdown to anticipate what components and subsystems exist for the device to perform its functions. The hands-on process took place in a maker space. Students were provided with basic hand tools. One of the tasks was to neatly place all the disassembled components on the table (see Figure 1). Then, students compared their anticipated diagram with the actual disassembled parts. They learned from taking notes of what was correct and what they had overlooked. During the process, we created a mini-roadshow by asking one student from each team to stay at their table and the rest of the class to walk to other tables to learn what steps other teams had carried out (see Figure 2). This activity facilitated discussions between students to discover new knowledge from peers. Furthermore, each student picked an area of interest (e.g., PCB design, user interface, materials, etc.) to conduct an independent study. The independent study required students to deeply dive into a piece of technology, a product, or a system and document their findings. This was a self-learning process, through which they were instructed to identify relationships between parts and recognize the interconnections of systems.

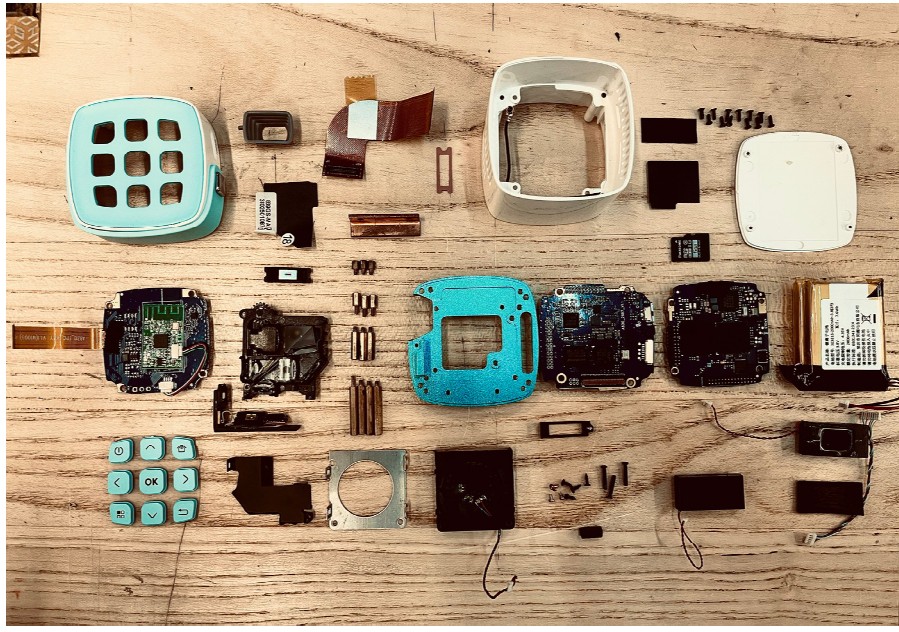

**Figure 1.** Students' submissions from the disassembling exercise.

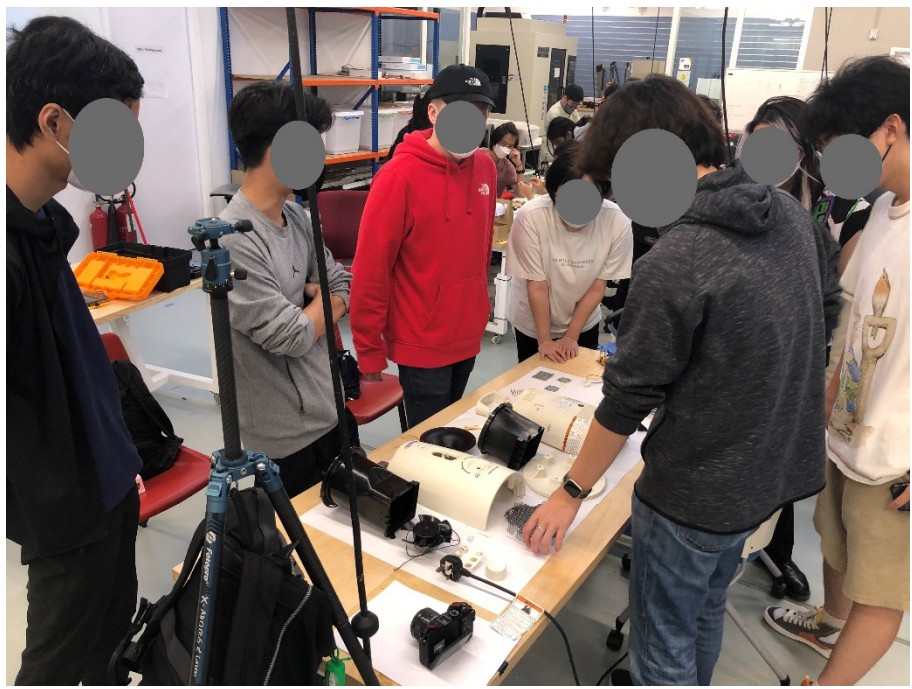

**Figure 2.** Students' interaction with the mini-roadshow.

Instead of specific topics, the last four weeks involve a mini-project that allows students to gather what they have learned in the previous modules and apply design- and systems-thinking processes to identify the problem they would like to solve. The project theme was "smart cities". This theme was chosen for several reasons:

- It is a contemporary topic of interest;
- It has a real-world impact;
- It is related to innovative technologies;
- It is interdisciplinary;
- It encompasses integrative systems ranging in size from infrastructural to microscopic.

The instructors briefly introduced the project theme and expected deliverables, then asked students to go through the design thinking process. This project phase relied heavily on student discussions and coaching by the instructors. Assessments were based on two main deliverables. Firstly, teams used storytelling to express the significance of their framed problem and proposed solution. Secondly, teams demonstrated their understanding of the complexity and connection of the different systems involved before, during, and after their solution takes place.

During the final week, students showcase their work in an exhibition format. The exhibition took place in a public concourse area on campus, where each team had a booth (a table and a display unit) to exhibit. Teams were assigned a timeslot, during which they had to station at their booth to answer questions from passersby (see Figure 3). Performance was evaluated by the instructors and via public feedback. Students were given the autonomy to decide how to demonstrate their design concept. Most teams used posters, prototypes, and pamphlets.

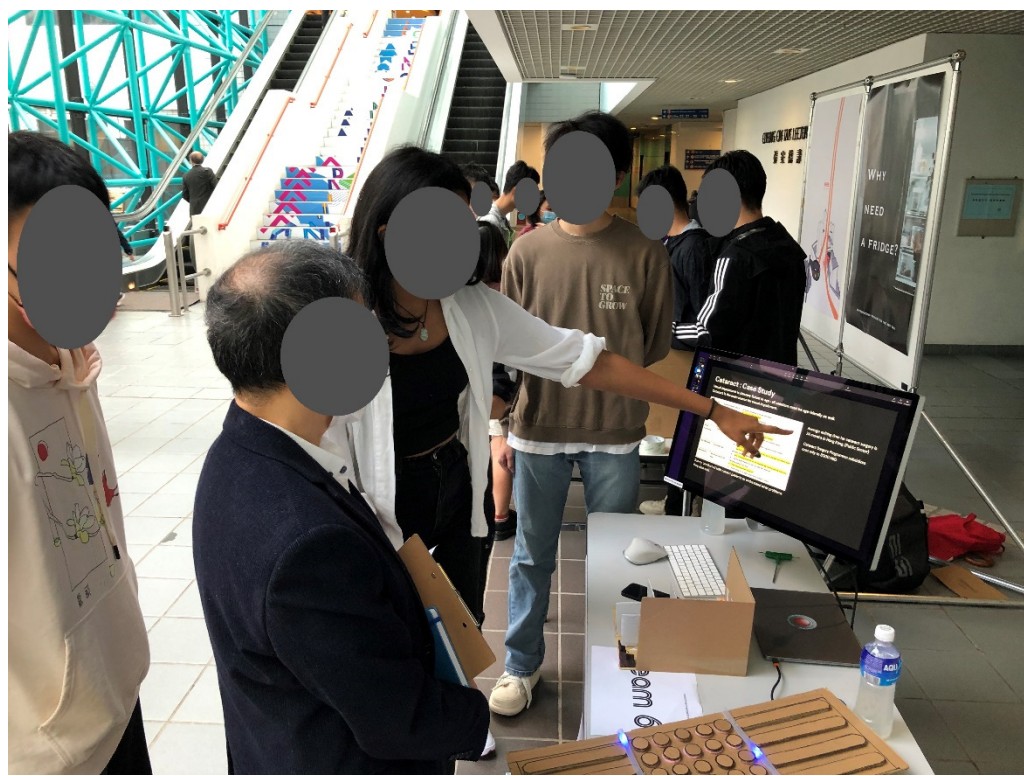

**Figure 3.** Smart city idea pitch in public on campus.

*5.2. Evaluation*

5.2.1. Course Feedback Surveys

Twenty-two students (69%) responded to the student feedback questionnaire. The course was positively received. Twenty-one respondents (95.4%) indicated 'agree' or "strongly agree" with the statement "The course materials and learning activities helped me achieve the course intended learning outcomes". And eighteen respondents (81.8%) indicated "agree" or "strongly agree" with the statement "Overall, I am satisfied with this course". In the open-ended question, students showed that they understand the value of this course lies in creating an atmosphere to inspire creativity. One student mentioned, "The course is interactive and successfully plants the seed of creativity and design in many otherwise engineering-minded students".

Another student appreciated the authentic learning experience and said, "I enjoyed the hands-on experiential learning that came with this course, it was a very practical start . . .".

At the beginning of the reflection exercise, students were reminded that participation is voluntary and that any response would not affect their course grades. Students used their electronic devices to provide the responses shown on the screen. Firstly, students were asked to recall things about the course. The majority of the responses indicated "field trip", which refers to the student–staff observation trip in the design phase.

Secondly, we asked students to share the most important knowledge, skills, or mindsets they have learned. Many responses were related to creativity, but only a few were related to systems or systems thinking. Some expressed gaining new insights into the engineering and technology discipline, "(I) see things with different perspectives, a new concept of design and technology, and expanded my eyesight in the technology field".

Thirdly, we asked a follow-up question on the strategies that helped their learning. Responses include hands-on group projects, group discussions, social relationships, and communicating with peers, teaching assistants, and professors. The fourth question was a 0–5 rating question asking how much students have improved in terms of the following aspects: creativity, design skills, systems thinking, project management, critical thinking,

communication, and technical knowledge. The mean score of most aspects was above the neutral score of 2.5, except for technical knowledge, which only scored 2.1 (see Table 2). While these results cannot infer conclusive claims about the effectiveness of this approach because of the lack of a pre-test and a control group, they form a constructive basis for follow-up interviews and the next course offering.

**Table 2.** Results from Mentimeter regarding students' perceived gains from the course.

| How Much Have You Gained after 3 Months? (0–5) | |
| --- | --- |
| Creativity | 2.8 |
| Design Skills | 3 |
| Systems Thinking | 3.3 |
| Project Management | 3.1 |
| Critical Thinking | 3.3 |
| Communication | 3.4 |
| Technical Knowledge | 2.1 |

5.2.2. Focus-Group Interviews

To understand how this learning experience affected students' mindset development, three focus-group interviews were conducted. Each group consisted of five participants, representing half of the class. The interviews consist of two parts. The first part asked students about their experience following the class activities in chronological sequence. The second part was about whether the course had helped to develop their mindsets.

Echoing results from the course feedback surveys, most students responded they enjoyed the student–staff observation trip and proposed that similar activities could be organized. Some students thought the tour in the city could inspire their design ideas, which also created an occasion for connecting and interacting with their peers and staff. A student said, "I didn't expect professors to be so friendly. We enjoyed the field trip very much. We learned a lot from him".

Regarding understanding systems, some students expressed this was an adequate introduction to engineering tools in a maker space and an opportunity to do something practical. The exercises allow them to see the delicacy of how products are designed, revealing smart design decisions in the manufacturing, packaging, or consuming processes of products. An example response was as follows: "I like the hands-on part because I'm interested in learning more about engineering". However, students who have had STEM or D&T classes during pre-tertiary education may find it less beneficial in the learning process. Another response was as follows: "The dissembling process is similar to the D & T (design and technology) lesson at my high school. I'm quite familiar with the tools".

Regarding the final project about smart cities, positive feedback was received on the idea of having an authentic scenario to work on, but negative comments on several aspects were reported on the execution of this module. Firstly, students generally felt that they did not have sufficient time to go through the design thinking process in the last four weeks. Although much information could be found on the internet and resources were posted on the course learning management system, students expressed difficulties in digesting the information and transforming it into a manageable project scope for them to work on. Some students felt confused about the expectations of their deliverables. They said that it was not clear how feasible and complete their proposed solutions needed to be. One comment received was as follows: "It's like the course teaches us to find the right problem. But in the smart city project, we were forced to come up with solutions in a very short time. Feels like a hackathon instead".

Overall, the positive comments can be summarized in three aspects. First is the project-based team learning environment: it brought authenticity to the experience and kept one-way lectures to a minimum. The second aspect is the continuing interaction between students and instructors throughout the course. A student commented, "The most



valuable thing I gained is friendship. I will probably do projects with these students in the next 4 years, this is good for icebreaking".

Students value social presence as part of a key component of an introductory course. Lastly, some students appreciated the flexibility given in their assignment submissions. The instructors emphasized that students should be creative when submitting their work or presenting in class, resulting in various kinds of media being used for the same assignment. For example, students were asked to use a journal to document and illustrate interesting observations of their field trip. The range of submissions included reports, diaries, video clips, cartoons, and app interfaces. A student mentioned, "I like how we can choose our own format to submit assignments".

The negative comments focused on the heavy workload and a lack of coherence between topics and modules. When students were asked to compare the workload to the courses they took during the semester, they generally agreed that the time and effort required for this course were relatively high. A student expressed, "One major issue with the course is the loading of assignments. We think that this is because of the large number of invited speakers". This statement is tied to the next issue about the transition from topic to topic. As explained earlier, the course used three learning theories to formulate activities and mini-projects in three modules. Two major criticisms were received. First is that students could not see the connection of the assignments between the design, systems, and integration phases. Instead, they felt overwhelmed with the due dates for submitting their work because the course had to move on to the next phase. The second critique was about scaffolding. Students felt it was difficult to apply the content covered in the first two modules to the final one. Although mindsets such as design- and systems-thinking should inherently be transferable, the assignments and projects had different focal points adjusted to the context of the topic presented by the invited speaker, and, thus, students may perceive the disconnection.

Based on the self-rated results on their perceived gain in various mindsets development (i.e., creative mindsets, habits of systems thinkers, and interdisciplinary awareness), students gave examples to elaborate on some of these aspects. While the learning modules had different focuses, the students' mindsets' development could be demonstrated in any parts of their learning experience. The participants were asked to take notice of any perceptual or behavioral changes over the course period. Specifically, the questions prompted were as follows: (1) Has this course changed your perspective on design and systems? If so, please explain how and give examples. (2) Has this course led to new habits or behaviors that are associated with creativity or systems thinking? If so, please explain how and give examples. (3) Has this course changed the way you work with others with different backgrounds? If so, please explain how and give examples. Their responses were categorized according to the principles of developing the mindsets stated in Section 4.2 (see Appendix A).

Regarding creative mindsets, some students acknowledged that problems can be identified if they become more observant of their surroundings. These problems may be related to a product, infrastructure, or a service. They find this learning experience useful to articulate their alertness, as it brought out the importance of being able to present these pain-points intellectually and emotionally. One student used one of the assignments as an example to support her view, ". . . I continue to expand my bugs list. I post it on my [social media] so I may come back to it one day to solve them". The bugs list was an exercise asking students to pay attention to details in their daily activities and spot anything that annoys them.

Many students mentioned team participation and learning to collaborate with their peers as one of the main developments they perceived. They realized that solving a problem as a team not only requires input from different discipline knowledge, but it is also important to acknowledge each team member's strengths and preferences. A student stated, "I value the experience to have a taste to integrate different discipline knowledge to

design a product". Another student added, "I learned that it is very important to know the strength and preference of my teammates and learn how to collaborate with them".

One foundation for being innovative is having an open mind. Instructors acted as facilitators in students' learning processes, and we asked questions to clarify their problem statements and design objectives and avoided providing any deterministic advice. A student said, "I remind myself that one problem can have many solutions. You just need to be creative about it".

Regarding the development of habits of systems thinkers, relatively few responses were received. There were no mentioned related to the principles of "thinking in systems" or "defining system boundaries". A few students claimed they developed a habit of self-learning. They understood self-exploration is necessary to be productive in design projects. A student remarked, "Most lectures were short and could only cover the basics, requires lots of individual studies. I discovered many things I am interested in further studying".

Regarding interdisciplinary awareness, the majority of the participants agreed they had gained new perspectives from working on interdisciplinary projects. An exemplary response was as follows: "I value the experience to have a taste to integrate different discipline knowledge to design a product".

Although this study did not include quantitative measurements to gauge the growth of mindsets in our students, the feedback is encouraging, as students have retained some practices as habits even after the course ended.

## 6. Discussion

This study illustrated the design, development, implementation, and evaluation of a pedagogical approach that aims to inspire design- and systems-thinking mindsets in first-year engineering students. Results from the course feedback surveys and the focus-group interviews are consistent.

In the student–staff observation trip, students could interact with teaching staff in an ill-structured environment. By understanding the staff's perspectives and thought processes, the field trip setting encourages students to process concepts from multiple angles. The only task in the field trip was to be observant; this leaves room for students to think and imagine. Students internalized the information they received and processed it into useful knowledge in their problem space. The environment encouraged students to see new opportunities and be curious about the world, which is advocated by Tom and David Kelly for developing creative mindsets [20].

The disassembling and reassembling exercise provided students with opportunities to be hands-on and look inside a product. This process not only helped students to visualize how a large system can be broken down into subsystems but also realized complex concepts such as emergence when subsystems synergize with one another to form new applications. The exercise also let students take ownership of the learning process, as instructors played the role of facilitators to guide and encourage them to discover. Previous studies integrating constructionism theory principles have found similar success (e.g., [32,33]).

Although polar responses were received regarding the final project, negative comments largely focused on the heavy workload. This can be improved if the scope of the project is adjusted and the project theme is announced earlier so that students can begin their independent studies earlier in the semester. Students worked in teams to investigate the possibilities and issues of technology in the smart city project. The process of having individual students conduct research in a particular area of interest and then share their findings in teams has helped them to raise their interdisciplinary awareness [34]. The project deliverables were evidence of collective thinking among team members. These are signs of distributed cognition [35]. Collaborative groups construct knowledge by collectively developing a mutual understanding, collaborating in the creation of innovative ideas, and engaging in interaction within a common problem area [36].

## 7. Conclusions

Our work contributes to the field as an attempt to design a new pedagogy for a mindset-focused introductory course. We argue that preparing creative mindsets, habits of a systems thinker, and interdisciplinary awareness are paramount to any engineering student in design- or systems-thinking curricula. Course activities were designed to minimize standardization and boundaries while promoting student autonomy and challenging assumptions. Findings indicated that students were positive about the learning experiences and showed perceived development in creative thinking, systems thinking, and interdisciplinary awareness.

There are a few limitations of this study that should be addressed in future work. From a methodological perspective, the current study mainly presented qualitative findings from course reflection and interviews. Validated questionnaires and tests shall be developed for triangulation and to provide quantitative evidence of students' mindset development. Further, the retention of mindsets can be elicited through longitudinal studies. From a practical perspective, the current format of some class activities requires a large number of teaching resources, including labor and project materials, making the class difficult to scale. The students' feedback from this study will form the basis for improvement in the next iteration.

Further refinement of the learning activities and assessments is necessary before achieving a comprehensive pedagogical design. More work needs to be carried out to ensure that the proposed approach is fully optimized. Given the extensive discourse on the current disruptions in higher education, this paper posits a project-based approach that may offer some insights into the sustainability of higher education.

**Author Contributions:** Conceptualization, J.K.L.L., D.T.K.N. and C.-Y.T.; methodology, J.K.L.L. and D.T.K.N.; validation, J.K.L.L., D.T.K.N. and C.-Y.T.; formal analysis, J.K.L.L. and D.T.K.N.; investigation, J.K.L.L. and C.-Y.T.; writing—original draft preparation, J.K.L.L.; writing—review and editing, J.K.L.L.; visualization, J.K.L.L. All authors have read and agreed to the published version of the manuscript.

**Funding:** This research received no external funding.

**Informed Consent Statement:** Informed consent was obtained from all subjects involved in the study.

**Data Availability Statement:** Data available on request due to privacy and ethical restrictions.

**Conflicts of Interest:** The authors declare no conflict of interest.

## Appendix A

Student Feedback Questionnaire
Closed-ended questions

Q1.  The course materials and learning activities helped me achieve the course intended learning outcomes.
Q2.  Assessment tasks were designed to determine the extent to which I had achieved the course intended learning outcomes.
Q3.  This course was academically challenging.
Q4.  The instructor stimulated my interest in this subject and encouraged me to think.
Q5.  The instructor was readily available to answer questions and support students.
Q6.  The instructor provided helpful and timely feedback on my performance (including, where applicable, in assessment tasks, tests, presentations, and projects).
Q7.  Overall, I am satisfied with the instructor's teaching.
Q8.  Overall, I am satisfied with this course.

Open-ended questions

Q9.  Please provide any further comments about the course.

Course Reflection Exercise

Q1.  What are some memorable class activities?
Q2.  What was the most important/valuable thing(s) you learned?
Q3.  Through what ways did you learn them?
Q4.  In terms of mindsets for integrative systems and design, how much have you gained after this course?

**Table A1.** Exemplary responses from focus-group interviews categorized by mindsets.

| Mindsets | Principles | Exemplary Response |
|---|---|---|
| Creative potential | Embrace empathy | "... I continue to expand my bugs list. I post it on my [social media] so I may come back to it one day to solve them". |
| | Embrace ambiguity | "I didn't expect so many sudden presentations. Now I feel speaking in public normal and safe". |
| | Embrace failure | nil |
| | Foster collaboration | "The course has many interactive activities and encourage team participation" |
| | | "I learned that it is very important to know the strength and preference of my teammates and learn how to collaborate with them". |
| | Encourage divergent thinking | "I remind myself that one problem can have many solutions. You just need to be creative about it". |
| | Practice reframing | "having to define the problem by ourselves was daunting, but it was a brand new experience for me" |
| | Cultivate a bias towards action | "hands-on experiences such as storytelling in every project and assignment to keep us in mind and develop the habits". |
| Habits of systems thinkers | Recognize interconnections | "the habit of really sketching out my thoughts is something I have not done in the past" |
| | Think in systems | nil |
| | Define system boundaries | nil |
| | Foster continuous learning, Foster a growth mindset | "Most lectures were short and could only cover the basics, requires lots of individual studies. I discovered many things I am interested in further studying". |
| Interdisciplinary awareness | | "I got to meet so many new people with so many different perspectives through this course, just looking at the high level of competency from my peers helped me, especially the field trip and disassembly exercise" |
| | | "I value the experience to have a taste to integrate different discipline knowledge to design a product" |
| | | "To see things with different perspectives, a new concept of design and technology. This course expanded my eyesight in technology field" |

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
