# Peer review of "A Primer for Design and Systems Thinkers: A First-Year Engineering Course for Mindset Development"

_education, doi:10.3390/educsci14010086_

Round 1

Reviewer 1 Report

Comments and Suggestions for Authors

Many thanks on the opportunity to review your current submission regarding the first-year engineering course for mindset development. First year engineering courses are a often a passion project, at times not receiving the same care as later courses in the curriculum. The retention impact on students from a properly developed first year course is significant, so it was with great anticipation that I approached your manuscript. 

In general, many of the intentions, goals, and ideas are well developed and appropriate for Education Sciences. Unfortunately, many of those promises feel unrealized and unfulfilled as the report meanders through a series of good intentions that fall relatively flat. These goals and potential significance are outlined at the end of the first section, specifically 'First, few studies on design thinking pedagogies center on nurturing students' mindsets for integrative systems and design. Second, this paper shares the experiences and good practices...Third this study provides empirical information on students' development in their values and mindsets...'

Overall, while the manuscript suffers from some grammatical issues, the largest issues are in the information provided and the missed intentions. The mindset information in section 2.3 is dramatically insufficient and does not provide the necessary framework to motivate the course design. The three learning theories in section 2.2 are not well developed in their discussion and only cursory information is provided to connect constructionism, distributed cognition, and cognitive apprenticeship to the cursory descriptions of activities in the course itself. 

From the perspective of section 3, there is no IRB or ethical information provided regarding the participants. Often, in first-year courses there are students not yet of legal age to provide permission to participate - another concern not addressed. Additionally, it is usually not advisable for instructors of the course to act as researchers; there are far too many biasing and lack of perspective issues. There were no details regarding the number of students in the course, no information regarding how the focus group participants were chosen, no information regarding how the students were protected if they chose not to participate in the study. 

In section 4, the course design was reasonably discussed, however there are no details regarding how the activities fit within the framework semi-described nor how they are intended to drive the mindsets previously suggested. The learning objectives presented at the end of section 4.1 seem to be vague and missing definitive transparent explanations. 

Section 5, describing the implementation, suffers from a complete lack of details regarding the expected time on task for the students, motivations, whether or not the course is required, how many classes there are, why the choices made were implemented (e.g. changing teams for each module when it commonly understood that a single team affords highest level of psychological safety and learning), etc. There is just no tie from the activities described back to the theory and supposed mindset development. The first module has none of the exercises shared, just that they 'represent the training on different dimensions such as flexibility, elaborateness...'. The activities in the design module do not tie back to how they demonstrate the appreciation of design, nor does it describe with sufficient depth the artifacts created. The reverse engineering activity in the systems module suggests 'independent study' but there is no further explanation.

Section 5.2 shares some of the evaluations, but the numbers provided seem to be misleading and not well explained. Further, there is no description of the full mindset development framework provided in the manuscript, so the focus groups trying to explore how the experiences affected that development are more around the course development and less around the actual mindset changes of the students.

Again, these are just the general sweeping statements surrounding this manuscript. There is great potential in the idea and the work done, however in it's current state this is more of an experience report than any sort of empirical research study. 

I look forward to the opportunity to help support the author(s) as they move forward, and many thanks for the chance to be of service!

Comments on the Quality of English Language

There are some grammar issues throughout, especially with regard to tenses, verb forms, pluralization, and preposition usage.

Reviewer 2 Report

Comments and Suggestions for Authors

The introduction and literature review are well written and researched. I was excitedly anticipating learning new course techniques and assessment strategies for systems thinking and design skills. However, I was disappointed that the data collected did not align with the literature cited. The data is basically a standard course evaluation (course workload, teacher effectiveness) rather than an assessment of the learning interventions employed.

It is not clear in the paper if IRB approval was received for the research. How many students in the course consented to participate in the study? Did their demographics match the overall demographics of the class?

Numbers on Figure 5 graph are hard to read. Does the shape of the colors behind the data bars relate to student responses? More description of this graph would be helpful.

Comments on the Quality of English Language

Overall paper is well written. There are a few very minor grammatical errors that would benefit from a re-read.

Round 2

Reviewer 1 Report

Comments and Suggestions for Authors

Many thanks to the author(s) for their detailed and considerate integration of previous suggestions into their revised manuscript. After a thorough reading of the current version, I have no further suggestions for additional consideration and/or improvement.

Thanks again for your efforts and I hope you and yours have a very happy 2024!

Reviewer 2 Report

Comments and Suggestions for Authors

Thank you for addressing my concerns. The updated draft is much improved.